# Industry Image Classification Based on Stochastic Configuration Networks and Multi-Scale Feature Analysis

**DOI:** 10.3390/s24154798

**Published:** 2024-07-24

**Authors:** Qinxia Wang, Dandan Liu, Hao Tian, Yongpeng Qin, Difei Zhao

**Affiliations:** 1Artificial Intelligence Research Institute, China University of Mining and Technology, Xuzhou 221116, China; 2Sunyueqi Honors College, China University of Mining and Technology, Xuzhou 221116, China; dandan.liu@cumt.edu.cn (D.L.); 08212823@cumt.edu.cn (H.T.); 05211866@cumt.edu.cn (Y.Q.)

**Keywords:** image classification, multi-scale analysis, stochastic configuration networks, feature extraction

## Abstract

For industry image data, this paper proposes an image classification method based on stochastic configuration networks and multi-scale feature extraction. The multi-scale features are extracted from images of different scales using deep 2DSCN, and the hidden features of multiple layers are also connected together to obtain more informational features. The integrated features are fed into SCNs to learn a classifier which improves the recognition rate for different categories. In the experiments, a handwritten digit database and an industry hot-rolled steel strip database are used, and the comparison results demonstrate the proposed method can effectively improve the classification accuracy.

## 1. Introduction

With the development of computer technology, deep learning methods have been widely used for image recognition and classification [1]. Since demonstrating their strong feature representation ability, CNNs with various structures have been developed [2,3,4,5,6] and widely used in image processing and analysis. For the industry image classification task, Masci et al. [7] presented a classification approach based on Max-Pooling CNNs with the features directly extracted from the pixel representation of the steel defect images. Lee et al. [8] proposed a classification method based on CNNs and class activation maps to implement a fast decision-making process. Chen et al. [9] proposed to combine three deep CNN models trained individually, and the average strategy was used to obtain defect classification. Konovalenko et al. [10] proposed a classifier based on two deep residual neural networks in which the hyper-parameters of the optimal model were selected through various investigations. Li et al. [11] proposed a CNN-T model by merging the CNN and Transformer encoder, and obtained an improvement in classification accuracy compared with the pure CNN. Feng et al. [12] introduced a ResNet50 classifier with two additional FcaNet and convolutional block attention modules to deal with steel surface defect data. To deal with the image classification problem, CNNs with different structures that consist of various modules have been developed and obtained better classification accuracy. However, in order to obtain the optimal model, the selection of hyper-parameters in the back-propagation algorithm requires a lot of prior work.

Random neural networks have also been developed for data analysis modeling; the weights and bias of the hidden layers are randomly assigned based on the randomized algorithm [13]. The feed-forward neural networks with random weights (NNRWs) was proposed in [14]. By adding the direct link between the input layer and output layer, Pao et al. [15,16,17] proposed the random vector functional link (RVFL) neural network. A stack RVFL was introduced in [18] using the negative correlation learning strategy. Lu et al. [19] proposed an image data recognition method using two dimensional neural networks with random weights (2D-NNRWs). The stochastic configuration networks (SCNs) in [20] were proposed to improve the learning process by introducing a supervised mechanism. Ensemble methods using base SCN models were introduced in [21,22] to deal with large-scale modeling. The base SCN model was expanded by using multiple hidden layers, and a deep network framework was built in [23,24]. For matrix data, a two dimensional stochastic configuration network (2DSCN) [25] has been developed, which has advantages in image data analytics. On this basis, Li et al. [26] proposed an improved SCN with vision patch fusion, which improves the network’s feature representation ability by extracting randomly fused image features from three-channel images. Li et al. [27] proposed to generate a different convolution kernel with physical meaning using a supervised learning mechanism, and a deep convolutional neural network was constructed for working condition recognition.

In this paper, stochastic configuration networks are used in the proposed image classification method. For feature extraction, compared with a single hidden layer, SCNs with deep structures can extract image features with low computing time. Moreover, because of randomly assigned weights and bias in the stochastic configuration algorithm, images of different scales are used to make up a loss of image details when constructing the deep networks. These multi-scale features are fused through a fully connected layer, and the integrated features are used to build a classifier based on SCNs to complete the identification of different categories. The main contributions are summarized as follows:(1)Designed an image classification framework based on SCNs for extracting features from multi-scale images.(2)Investigated the influence of different scales’ training data and different network structures on feature extraction results.(3)Demonstrated the advantage of the proposed randomized learning method in steel surface defect data.

The remainder of this paper is organized as follows. Section 2 overviews the stochastic configuration networks. Section 3 details the proposed method with deep feature extraction SCNs and a learned classifier. Section 4 elaborates on the experiments and discusses the experimental results. Finally, the conclusion is presented in Section 5.

## 2. Stochastic Configuration Networks

As improved randomized networks, SCNs were proposed in [20] under a supervised mechanism. Based on the stochastic configuration algorithm, the weights and biases of the hidden layer are gradually configured until an SCN is built. The modeling process of the SCN model with a single hidden layer can be described as follows:(1)Given a training dataset {X,T} with N samples, where $X=\{x_{1},x_{2},\ldots ,x_{N}\}$, $x_{i}\in R^{d}$, $T=\{t_{1},t_{2},\ldots ,t_{N}\}$, $t_{i}\in R^{m}$.(2)Suppose the SCN has been configured with L−1 hidden nodes, the output can be calculated by
(1)$$\begin{array}{c} f_{L-1}\left(X\right)=\sum_{l=1}^{L-1}\beta_{l}g_{l}\left(w_{l}^{T}X+b_{l}\right),L=1,2,\ldots .\left(f_{0}=0\right) \end{array}$$
where $w_{l}\in R^{d},b_{l}\in R$ are the weights and bias, $\beta_{l}=\left[\beta_{l,1},\beta_{l,2},\ldots ,\beta_{l,m}\right]$ is the weight of the output layer and *g* is an activation function. The current residual error is
(2)$$\begin{array}{c} e_{L-1}=T-f_{L-1}\left(X\right)=\left[e_{L-1,1}\left(X\right),e_{L-1,2}\left(X\right),\ldots ,e_{L-1,m}\left(X\right)\right]. \end{array}$$(3)Suppose the error $e_{L-1}$ does not reach the stop condition, then a new node should be added and the weight and bias $w_{L},b_{L}$ are configured to calculate the hidden output
(3)$$\begin{array}{c} h_{L}\left(X\right)=g_{L}\left(w_{L}^{T}X+b_{L}\right). \end{array}$$

Denote a set of variables $\xi_{L,q},q=1,2,\ldots ,m$ as follows:(4)$$\begin{array}{c} \xi_{L,q}=\frac{{\left(e_{L-1,q}{\left(X\right)}^{T}\cdot h_{L}\left(X\right)\right)}^{2}}{h_{L}{\left(X\right)}^{T}\cdot h_{L}\left(X\right)}-\left(1-r-\mu_{L}\right)e_{L-1,q}{\left(X\right)}^{T}e_{L-1,q}\left(X\right) \end{array}$$
where 0<r<1, $\mu_{L}$ is a sequence of real numbers and $0<\mu_{L}\leq 1-r,\lim_{L\to \infty }\mu_{L}=0$.

(4)The configured weights should satisfy the following constraint condition
(5)$$\begin{array}{c} \xi_{L,q}\geq 0,\;q=1,2,\ldots ,m. \end{array}$$

Based on the SC-I algorithm in [20], a set of candidate parameters $w_{L}^{n},b_{L}^{n}$ are randomly assigned under the condition Equation (Equation 5). The corresponding parameters are selected as the final values when $\sum_{q=1}^{m}\xi_{L,q}$ obtains the maximum value.

(5)Calculate the output weight β. For the configured and fixed hidden weights and bias, the hidden feature matrix is $H=[h_{1},h_{2},\ldots ,h_{L}]$ with $h_{l}=g\left(w_{l}X+b_{l}\right),l=1,2,\ldots ,L$. Then, the output weight $\beta =[\beta_{1},\beta_{2},\ldots ,\beta_{L}]$ can be calculated
(6)$$\begin{array}{c} \beta^{★}=\arg\min_{\beta }||f-\sum_{l=1}^{L}\beta_{l}h_{l}{\left|\right|}_{2}^{2}. \end{array}$$

In the stochastic configuration algorithm, the hidden layer weights $w_{l},l=1,2,\ldots ,L$ are firstly taken from a certain distribution (e.g., Uniform distribution or Gaussian distribution), and the bias $b_{l}$ can be calculated by
(7)$$\begin{array}{c} b_{l}=-w_{l}^{T}X^{★},\;l=1,2,\ldots ,L, \end{array}$$
where $X^{★}$ is randomly taken from the training dataset [28]. Then, the supervision mechanism (Equation 5) is employed to select proper weight values and add new hidden nodes. The deepSCN is an expansion of the SCN framework and more details about the theory and algorithm can be found in [23].

Similarly, for the image data, i.e., $x_{i}\in R^{d1\times d2}$, a 2DSCN model can be built under the supervised constraint Equation (Equation 5), and the output of the hidden node is
(8)$$\begin{array}{c} h_{l}\left(X\right)=g_{l}\left(u_{l}^{T}Xv_{l}+b_{l}\right),\;l=1,2,\ldots .,L, \end{array}$$
where $u_{l}\in R^{d1}$, $v_{l}\in R^{d2}$ are the hidden layer weight parameters and they are randomly assigned, and the bias is calculated via $b_{l}=-u_{l}^{T}X^{★}v_{l}$ with randomly sampled data $X^{★}$. The weight of the output layer $\beta_{l}$ is calculated by Equation (Equation 6), and the model output is
(9)$$\begin{array}{c} f_{L}\left(X\right)=\sum_{l=1}^{L}\beta_{l}h_{l}. \end{array}$$

Moreover, by combining the learning processes of a 2DSCN and deepSCN, the construction of a deep 2DSCN can be described as follows. When the structure of deep networks is in series, the output of the current layer is regarded as the input data of the next layer. Due to the stochastic configuration algorithm, the weights and bias of the hidden layers are randomly generated from a certain distribution and selected with the inequality condition (Equation 5) satisfied. The weight parameters of the first hidden layer and the following hidden layers are $u_{l},v_{l},b_{l}$ and $w_{k},b_{k}$; the corresponding hidden features are extracted to calculate the parameter β and establish the deep 2DSCN model.

## 3. Proposed Multi-Scale Image Classification Based on SCNs

In order to extract multi-scale information and improve the feature representation ability, an SCN-based classification method is proposed in this paper. The overview of the proposed method is shown in Figure 1, the multi-scale images are obtained by downsampling from the source image data, and the images of each scale are used to construct a deep SCN model for feature extraction, respectively. These extracted features are combined via a fully connected layer, and an SCN-based classifier is built to obtain the classification result.

### 3.1. Multi-Scale Feature Extraction Based on deepSCNs

During the pre-processing process, images of different scales are obtained from the original data by performing kernel convolution operators and downsampling. For an image *I*, the new image of a different scale can be obtained by
(10)$$\begin{array}{c} I_{new}=D_{p}\left[\;Conv\left(I,\;\sigma \right)\;\right], \end{array}$$
where σ is the Gaussian kernel and $D_{p}$ is the downsampling operator.

Given the training dataset {X,T}, where $X=\{x_{1},x_{2},\ldots ,x_{N}\}$, $x_{i}\in R^{d1\times d2}$ and target label $T=\{t_{1},t_{2},\ldots ,t_{N}\}$, $t_{i}\in R^{m}$. The Gaussian kernels $\sigma_{n}$ are used to obtain K sub-dataset $\{X^{n},T^{n}\},n=1,2,\ldots ,K$ by
(11)$$\begin{array}{c} X^{n}=D_{p}\left[\;Conv\left(X,\;\sigma_{n}\right)\;\right],\;n=1,2,\ldots ,K. \end{array}$$
where $\sigma_{n},n=1,2,\ldots ,K$ are different kernels and are used to obtain images of different scales. Since the label is unchanged, $T^{n}=T,n=1,2,\ldots ,K$.

For each sub-dataset $\left\{X^{n},T^{n}\right\}$, the deep SCN model is built for feature extraction. The detailed modeling process is described in Algorithm 1. Firstly, the images $X^{n}$ are used to extract the feature matrix $H_{1}$ of the first hidden layer, i.e.,
(12)$$\begin{array}{c} H_{1}\left(X^{n}\right)=\left[h_{1}\left(X^{n}\right),h_{2}\left(X^{n}\right),\ldots ,h_{L}\left(X^{n}\right)\right], \end{array}$$
where $h_{l}\left(X^{n}\right)=g\left(u_{l}^{T}X^{n}v_{l}+b_{l}\right)$. If the number of hidden nodes is equal to $L_{max}$ or the early stop condition is satisfied, the corresponding weights and bias $\left\{u_{l},v_{l},b_{l}\right\}$ are fixed, and the parameter learning of the next layer proceeds.

For the second hidden layer, since the network is connected in series, the input data are the feature vectors in the matrix $H_{1}\left(X^{n}\right)$ which is extracted from the layer above. For each feature vector, the weights and bias $w_{l},b_{l}$ are randomly assigned through the inequality constraint and the feature matrix $H_{2}\left(X^{n}\right)$ is calculated. The network structure is built step by step with the adding of hidden nodes until the stop condition is met.

Suppose the deep SCN model has M hidden layers, for each sub-dataset, the final feature matrix can be obtained by combining the hidden layer features,
(13)$$\begin{array}{c} H\left(X^{n}\right)=\left[H_{1}\left(X^{n}\right),H_{2}\left(X^{n}\right),\ldots ,H_{M}\left(X^{n}\right)\right]. \end{array}$$
**Algorithm 1:** Deep SCNs for images.
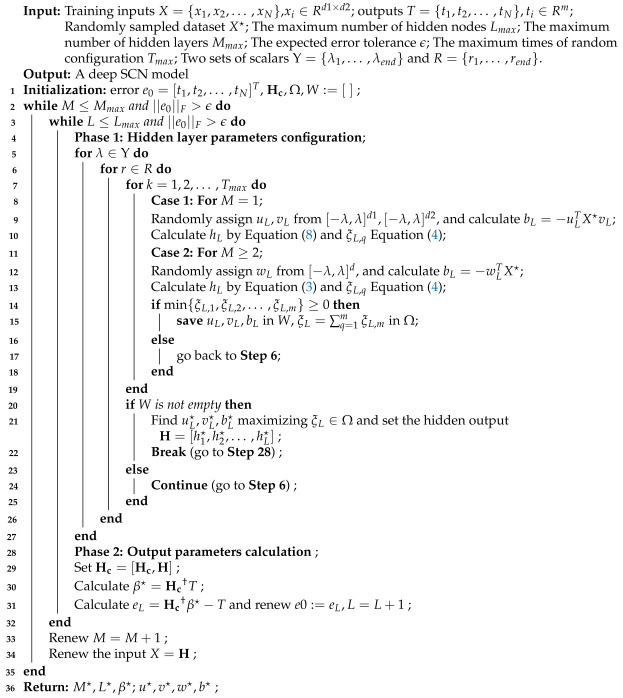



### 3.2. SCN-Based Classifier

For the multi-scale feature extraction, the deep SCN models are established by executing Algorithm 1 on each sub-dataset, and the corresponding feature matrix $H(X^{n})$ is calculated as the output.

A fully connected layer is used to fuse the features extracted from images of different scales; the fully connected feature vector can be calculated by
(14)$$\begin{array}{c} H=[H\left(X^{1}\right),H\left(X^{2}\right),\ldots ,H\left(X^{K}\right)]. \end{array}$$

Then, the feature data H are used for building the SCN classifier, and the classification results are obtained.

The proposed classification method consists of two stages: the multi-scale feature analysis and classifier learning. For multi-scale feature analysis, the sub-images of different scales are obtained by various convolution kernels operators. Image features are extracted from sub-images by building a deep SCN model, and in order to reduce the information loss among hidden layers, the hidden features of each layer are combined together. Then, the connected features are used to build the classification model, and an SCN-based classifier is learned to identify different categories.

## 4. Performance Evaluation

### 4.1. Experimental Setup

In this section, the Handwritten Digit Recognition CVL database [29] and the Northeastern University (NEU) surface defect database [30,31] are used for experimental comparison. The CVL database serves as a standard database for digit recognition and classification and is usually used for performance comparison among various algorithms. The NEU dataset has six kinds of typical surface defects of the hot-rolled steel strip, including rolled-in scale (RS), patches (Pa), crazing (Cr), pitted surface (PS), inclusion (In) and scratches (Sc), as shown in Figure 2. The dataset includes 1800 images with the size 299×299. Due to the small sample size, the deep neural network model is prone to overfitting. The window sampling method is applied to expand the number of samples to 10,000, and the new sample images are resized to 64×64. In the experiment, the image data are randomly sampled, with 80% as the training data and the remaining 20% as testing data.

In the proposed method, for the parameters used to build SCNs, we take the maximum number of candidate layers $M_{max}=5$, the training tolerance error ϵ=0.0001, the range of the hidden weights λ={0.5,1,5,30,50,100}, r={0.9,0.99,0.999,0.9999,0.99999}, the maximum number of candidate nodes $T_{max}=100$ and *g* is the sigmoid activation function. All the simulation experiments in this paper are performed by using the MATLAB R2023a on a computer with a 3.00 GHz CPU and 16 GB memory.

### 4.2. Results and Discussion

We first evaluate the classification results of deep SCNs with different numbers of hidden layers. For the CVL database, the number of hidden layers $M_{max}$ is set to {1,2,3,4,5} with $L_{max}=500$. Since the SCNs are built by incremental learning in Algorithm 1, the classification accuracy of different hidden nodes *L* is calculated. As shown in Figure 3, with the increase in hidden layer nodes, the classification accuracy of the training and testing process increases. For M=1,M=2,M=3, we find that the results of the test data are very close when L=500, while the computing time shown in Table 1 is completely different. For the NEU dataset, the number of hidden layers $M_{max}$ is set with $L_{max}=1000$, as shown in Figure 4. For the hidden layer M=2, the values of the classification result are close to that of M=1 before L=500; then, the values get higher with the increment in hidden nodes *L*. However, the curve values of M=4 and M=5 decrease and the classification effect becomes worse.

The comparison results of different *M* show that large *M* can extract more useful and reliable feature information for image representation, the built deep SCN model can obtain a good recognition rate, while the computing time is shorter. In addition, due to the limited number of training samples, the deep model may become overfitted. Different from the back-propagation algorithm, the deep network of the proposed method is built by the stochastic configuration algorithm; the modeling process is an incremental learning process, meaning it can adaptively adjust the network structure.

In order to analyze the influence of image size on modeling, the original image data are downsampled to obtain samples of different sizes. The original sizes of the images in the CVL database and the NEU dataset are 28×28 and 64×64, respectively. Since the CVL database consists of images with handwritten digits, the image size has a significant impact on molding. As shown in Figure 5, image samples of three different scales are used to build the SCN and deepSCN models. The comparison results show that the recognition rate gets higher as the image size becomes smaller, no matter whether using the SCN model or deepSCN model. The comparison results on the NUE dataset are shown in Figure 6. Different from the CVL dataset, the training recognition rate gets lower as the image size becomes smaller. With the addition of hidden nodes L, the recognition rate keeps increasing, and the model built with image data from scale 2 achieves high testing accuracy. In the deep network, the input of the current hidden layer is the extracted feature map from the above layer. The features extracted from scale 2 images can still be used to characterize the images, whereas the images from scale 3 have a loss of detail, resulting in low recognition rates. Moreover, the computing time is calculated and is shown in Table 2. In general, the larger the image, the longer the computing time. However, compared with the SCN model, the deepSCN model achieves a comparable recognition rate and is less time-consuming.

For the image classification task, the proposed method consists of different parts, including downsampling to obtain images of different scales, building the deep SCN model for feature extraction and building an SCN classifier. For the three parts, the ablation experiments are performed. As shown in Figure 7, the SCN and deepSCN methods are built with only one-scale images (i.e., the original image dataset). The ensemble-SCN and ensemble-deepSCN methods are built with images from multi-scales and their outputs are integrated into the final classification results. The proposed-SCN and the proposed-deepSCN methods are the proposed method in which the connected feature vectors are extracted by the SCN and deepSCN models.

As shown in Figure 7, because the training set only contains images from one scale, the recognition rate of the SCN and deepSCN methods is the lowest; see the yellow curves. With training images collected from multiple scales, the recognition rate of the ensemble-SCN and ensemble-deepSCN methods is greatly improved, while when the number of hidden nodes *L* exceeds 500, the recognition rate begins to decrease due to overfitting. The proposed method can solve the overfitting problem by adding an additional classifier to optimize the fully connected feature vector. Through the experimental comparison, the proposed-deepSCN method can obtain a good recognition accuracy, and due to the stochastic configuration algorithm, the building of networks with a deep structure takes less cost time than the single hidden layer structure.

In the experiment, the proposed method is compared with some randomized learning methods, such as the 2DNNRW- [19], 2DSCN- [25], deepSCN- [23], DNNE- [18] and SCNE [22]-based methods. Table 3 shows the recognition results of the comparison methods. It can be seen that the 2DNNRW-, 2DSCN- and deepSCN-based methods obtain low classification accuracy as the training data are constructed from single-scale images. The DNNE- and SCNE-based methods built with multi-scale images obtain higher accuracy along with the proposed method. In addition, when an additional classifier is added, the proposed method achieves a higher recognition rate. The comparison results show that the multi-scale feature representation and classifier of the proposed method can effectively improve the classification accuracy.

## 5. Conclusions

In this paper, an image classification method using SCNs is proposed. Due to its incremental learning algorithm, the network structure of deep SCNs can be adaptively constructed with the weight parameters of hidden layers randomly assigned. The experimental results demonstrate the advantage of the proposed method for solving classification problems, using the multi-scale extracted features, and an additional classifier can effectively improve classification accuracy. For future work, the weight configuration of the convolutional kernel can be discussed to optimize SCNs’ modeling on image data.

## Figures and Tables

**Figure 1 sensors-24-04798-f001:**
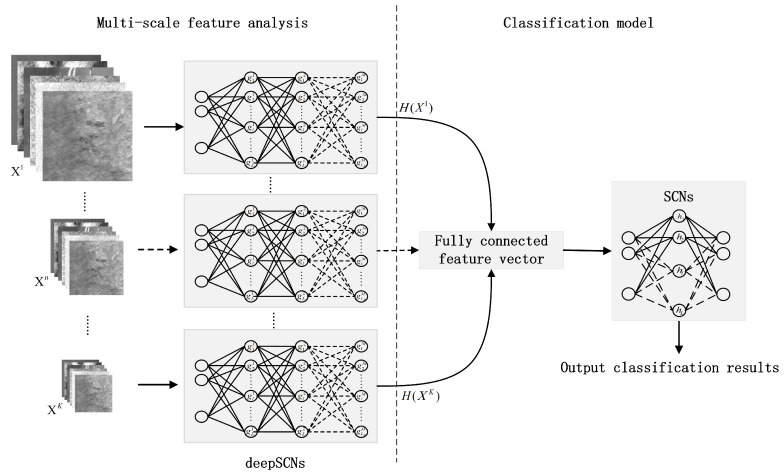
The overview of the proposed method.

**Figure 2 sensors-24-04798-f002:**
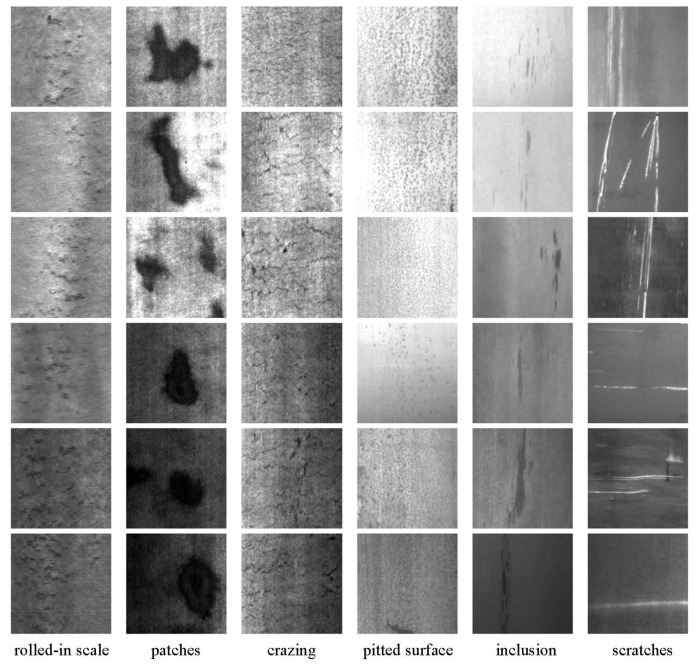
The NEU surface defect data.

**Figure 3 sensors-24-04798-f003:**
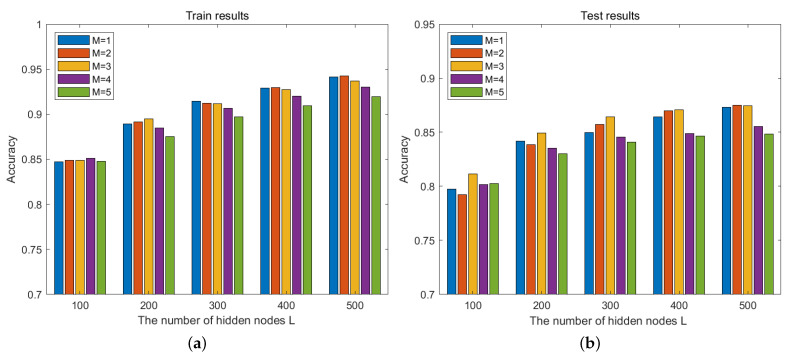
Recognition rate of different *M* on CVL database. (**a**) Training performance; (**b**) testing performance.

**Figure 4 sensors-24-04798-f004:**
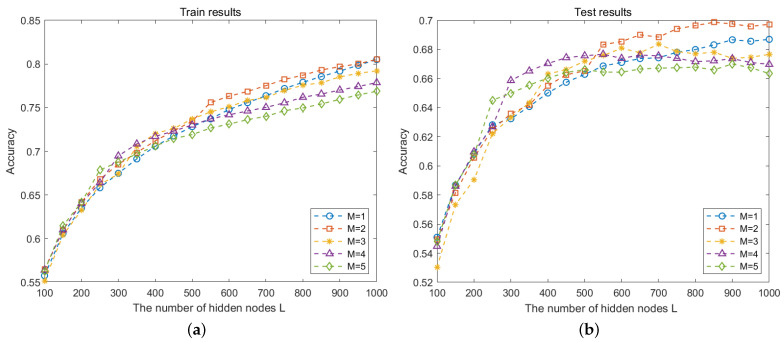
Recognition rate of different *M* on NEU database. (**a**) Training performance; (**b**) testing performance.

**Figure 5 sensors-24-04798-f005:**
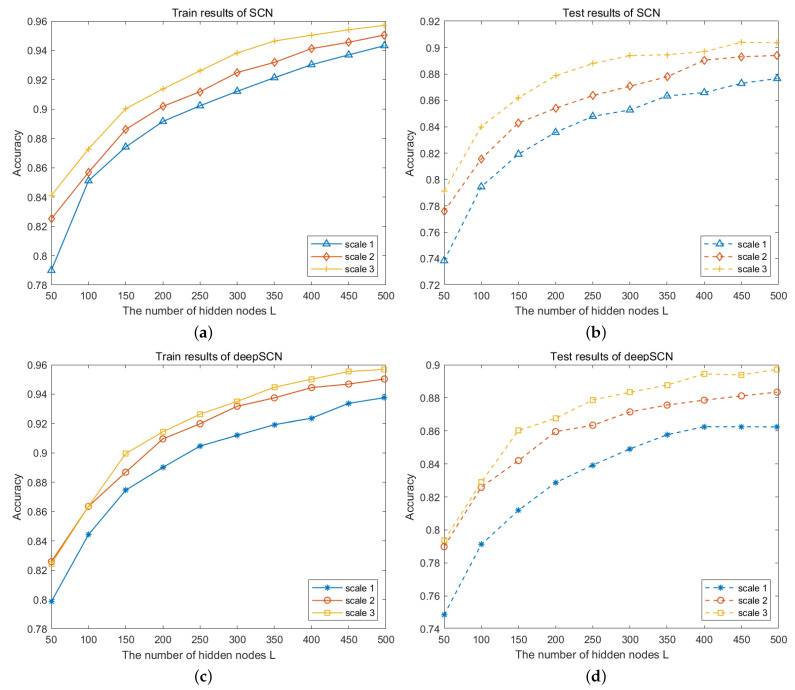
Performance comparison on images with different sizes on CVL database. (**a**,**b**) Training and testing results of the SCN model; (**c**,**d**) training and testing results of the deepSCN model.

**Figure 6 sensors-24-04798-f006:**
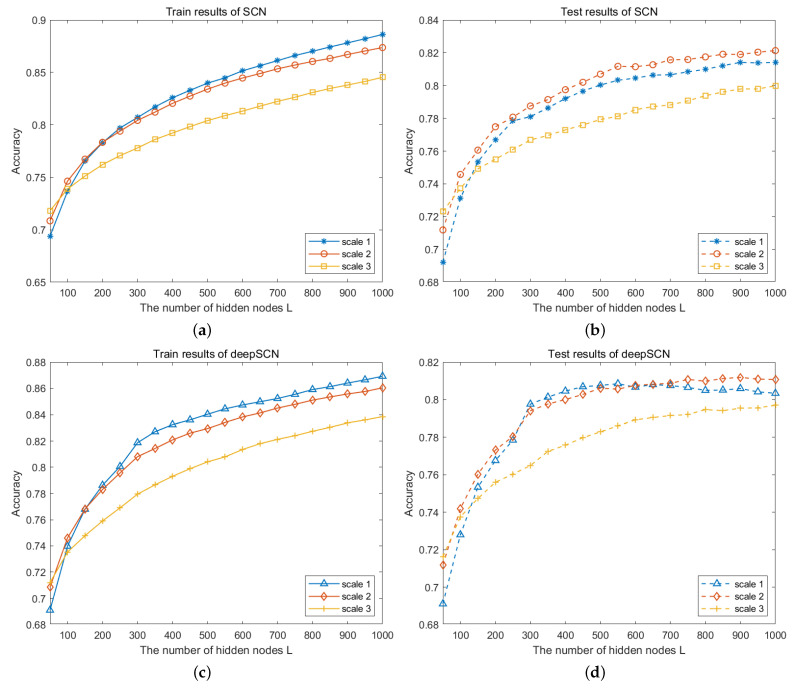
Performance comparison on images with different sizes on NEU database. (**a**,**b**) Training and testing results of the SCN model; (**c**,**d**) training and testing results of the deepSCN model.

**Figure 7 sensors-24-04798-f007:**
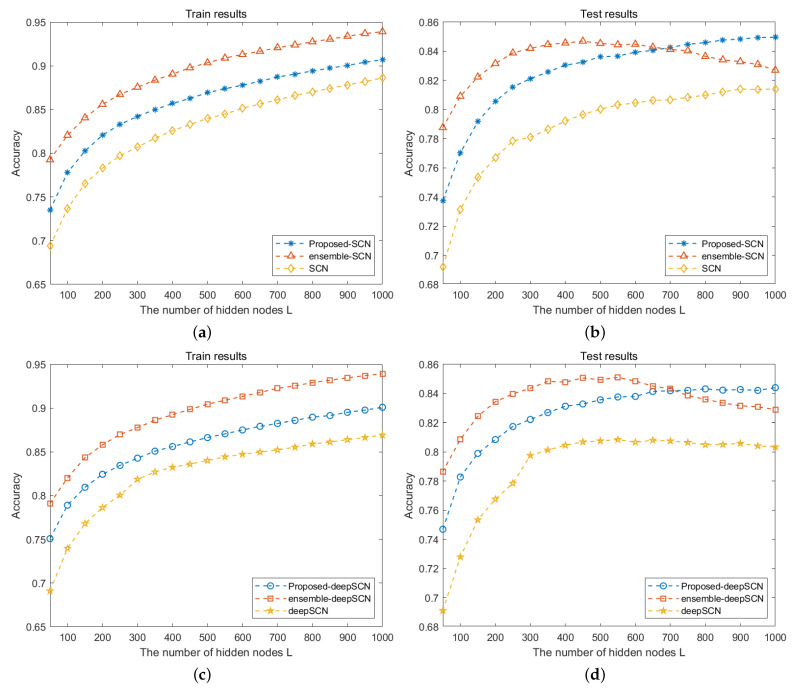
Performance comparison with different classifiers on NEU database. (**a**,**b**) Training and testing performance with SCN-based feature extraction; (**c**,**d**) training and testing performance with deepSCN-based feature extraction.

**Table 1 sensors-24-04798-t001:** Computing time of deep SCN model with different layers.

	M = 1	M = 2	M = 3	M = 4	M = 5
CVL	3004.3078	826.2544	417.0548	275.0342	198.1505
NEU	4084.9769	679.8829	386.1502	317.3123	161.1162

**Table 2 sensors-24-04798-t002:** Computing time of different image sizes.

	Scale	Image Size	deepSCN	SCN
CVL	scale 1	28 × 28	438.7538	4349.2825
	scale 2	14 × 14	383.0500	3823.0818
	scale 3	7 × 7	403.4073	3910.1134
SEU	scale 1	64 × 64	302.7245	4089.6932
	scale 2	32 × 32	240.2781	2924.6824
	scale 3	16 × 16	221.6883	2583.3933

**Table 3 sensors-24-04798-t003:** Comparison results on NEU dataset.

Model	Accuracy	Recall	Precision	F1 Score
2DNNRW	0.7691	0.6167	0.8864	0.7275
2DSCN	0.8168	0.6949	0.9190	0.7914
deepSCN	0.8125	0.6873	0.9170	0.7857
DNNE	0.7917	0.6556	0.9007	0.7589
SCNE	0.8538	0.7574	0.9384	0.8382
Proposed	0.8559	0.7606	0.9398	0.8407

## Data Availability

Data are contained within the article.

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
