# Peer review of "Industry Image Classification Based on Stochastic Configuration Networks and Multi-Scale Feature Analysis"

_sensors, 2024, doi:10.3390/s24154798_

Round 1

Reviewer 1 Report

Comments and Suggestions for Authors

This paper proposes a classification method based on stochastic configuration networks and multi-scale feature representation for industry image data, which contains two steps: 1) Images from different scales are used to build deep SCN models, and multi-scale features are extracted; 2) The extracted features are fused using a fully connected layer, and an SCN classifier is built to improve the recognition rate of categories. The experiments on a handwritten digit database and an industry hot-rolled steel strip database demonstrate the effectiveness of the proposed approach. This paper is well organized, and the topic looks interesting. 

I still have the following comments:

1) The author should review more works about multi-scale image classification since it is an important feature of this study.

2) Upon the equation (7), the abbreviation “SC” doesn’t have a full name. 

3) The Figure 7 is confusing. There are six lines in the figure. Are the “ensemble-SCN-Classifier” and “ensemble-deepSCN-Classifier” combined with the proposed networks? Or all the six classifiers are combined with the proposed networks? 

4) Are there any other methods for multi-scale industry image classification? If other different methods exist, authors should add related comparison experiment results.

Reviewer 2 Report

Comments and Suggestions for Authors

This paper presents a multiscale network based on SCNs design. To address the issue of insufficient sample size in the dataset, it propose extracting features at different scales using convolutional layers. These features are then separately fed into SCNs for extraction, followed by fusion through fully connected layers. The fused features are subsequently input into SCNs to obtain classification features.

Overall, this paper does not modify SCNs themselves; rather, it focuses on their stacked combination. Some more improvements on SCNs may be conducted and discussed.

Regarding the fusion of multiscale features, this paper only utilizes fully connected layers for fusion, which is considered overly simplistic and should be more carefully designed.

In terms of presentation, this paper devotes considerable space to detailing SCNs, which is unnecessary given the paper's focus should be on presenting its own contributions as comprehensively as possible.

Some prior works involving multi-scale feature analysis are lacked, either hand-crafted and deep learning based ones, e.g., M2FN: A Multilayer and Multiattention Fusion Network for Remote Sensing Image Scene Classification –GRSL22, and LETRIST: Locally Encoded Transform Feature Histogram for Rotation-Invariant Texture Classification –TCSVT2018.

In terms of experiments, this paper conducts extensive ablative experiments; however, it lacks comparative experiments to demonstrate the superiority of the proposed method.

Finally, the paper mentions that the primary role of multiscale features is data augmentation, but traditional methods such as flipping and cropping were not attempted. Combining these methods could enhance the approach presented in this paper.

The writing needs to be improved, e.g, the abstract part contains a very long yet wrong sentence (…firstly…then…and…).

Comments on the Quality of English Language

can be improved

Reviewer 3 Report

Comments and Suggestions for Authors

The article requires major revision.

1. I suggest that the authors further clarify the method proposed in the introduction section, as it appears that they have reused the same paragraph as in the abstract.

2. It is essential to add a paragraph at the end of the introduction section to highlight the contributions of this work.

3. I suggest adding a section entitled "Related Work" in which the authors highlight the advantages of the proposed method over existing methods. It would be preferable to present this section in tabular form.

4 . I have a major ambiguity regarding your method, which is based on SCNs with a multi-scale feature representation.

-Does the use of the multi-scale feature representation increase the efficiency of the classifier?

-I suggest the authors add a comparison between the proposed method with the multiscale representation and without the multiscale representation.

5 . The simulations used are based on a database containing grayscale images. Is the method also applicable to color images?

6- in the "Performance Evaluation" section I have some ambiguity

-What types of image augmentation techniques were used to expand the samples to 10,000?

-How was the image size of 64 × 64 determined to be optimal for this study?

-Why was the 80% training and 20% testing dataset split chosen, and how does it impact the performance of the model?

-Were there any specific criteria or methods used for the random selection of training and testing datasets?

-How does the dataset split ensure that the training and testing sets are representative and unbiased?

7-The references section contains mainly older sources, whereas an article scheduled for publication in 2024 should include references from the years 2021, 2022 and 2023.

Doi.org/10.3390/e23010056

Doi.org/10.3390/s20154191

DOI: 10.1016/j.bspc.2023.105128

Doi.org/10.3390/app13052878

DOI: 10.1007/s11042-023-15582-9

8- Plagiarism rates should be kept to a minimum

Comments on the Quality of English Language

must be improved

Round 2

Reviewer 2 Report

Comments and Suggestions for Authors

This paper only uses fully connected layers to fuse multiscale features, which lacks enough novelty. Some related multiscale works such as HRNet and conventional methods should be discussed.

Comments on the Quality of English Language

shoud be improved

Author Response

Comments and Suggestions for Authors

This paper only uses fully connected layers to fuse multiscale features, which lacks enough novelty. Some related multiscale works such as HRNet and conventional methods should be discussed.

Response: Due to the stochastic configuration algorithm, the output of a sample ( an image) in the hidden layer is a vector. For each sub-dataset  (n = 1, 2,..., K ), the feature matrix is , where N is the total number of training samples and L is the number of hidden nodes. As shown in Table 2, the proposed method obtains a recognition accuracy of 85.59%. For conventional fusion methods, the weight average fusion method has been tested and obtained an accuracy of 84.91%.

Due to the size of the output hidden matrices is same, the fusion module of the HRNet cannot  be performed in the proposed method. In our further work, the kernel SCNs will be discussed which can use the fusion module in HRNet or other convolutional networks.

Reviewer 3 Report

Comments and Suggestions for Authors

After a thorough review of the revised paper, I find that the authors have addressed all the reviewers' comments. Therefore, I recommend the publication of the paper in its current form.

Comments on the Quality of English Language

Moderate editing of the English language is needed.

Author Response

The English language has been edited.